# Conditioning on Time is All You Need for Synthetic Survival Data Generation

## Abstract

Synthetic data generation holds considerable promise, offering avenues to enhance privacy, fairness, and data accessibility. Despite the availability of various methods for generating synthetic tabular data, challenges persist, particularly in specialized applications such as survival analysis. One significant obstacle in survival data generation is censoring, which manifests as not knowing the precise timing of observed (target) events for certain instances. Existing methods face difficulties in accurately reproducing the real distribution of event times for both observed (uncensored) events and censored events, *i.e.*, the generated event-time distributions do not accurately match the underlying distributions of the real data. So motivated, we propose a simple paradigm to produce synthetic survival data by generating covariates conditioned on event times (and censoring indicators), thus allowing one to reuse existing conditional generative models for tabular data without significant computational overhead, and without making assumptions about the (usually unknown) generation mechanism underlying censoring. We evaluate this method via extensive experiments on real-world datasets. Our methodology outperforms multiple competitive baselines at generating survival data, while improving the performance of downstream survival models trained on it and tested on real data. Importantly, our approach delivers these improvements without compromising data privacy, offering an effective solution for synthetic survival data generation.

## 1 Introduction

Synthetic data generation is the process of creating artificial data that mimics the statistical properties and patterns of real-world data. This technique has gained significant importance in various machine learning settings including data privacy and data augmentation (Jordon et al., 2022). The primary motivation behind synthetic data generation is to address challenges associated with limited availability, privacy concerns, or imbalance in distributions often prevalent in real-world data (Zhang et al., 2017; Wang et al., 2021). For instance, researchers, practitioners and organizations could train and evaluate machine learning models by leveraging synthetic data without compromising sensitive or proprietary information. Further, synthetic data can augment existing datasets, enabling more robust and generalized model performance. Alternatively, it can protect data privacy by providing a means to share and exchange data without revealing sensitive information, facilitating collaboration and research across different domains (de Benedetti et al., 2020).

Survival analysis, also known as time-to-event analysis, is a family of statistical methods used to analyze and model the time until the occurrence of a specific event (or outcome) of interest. These methods are widely employed in various fields, including biomedical research, operations research, engineering, economics, and social sciences (Kaso et al., 2022; Lillelund et al., 2023; Danacica & Babucea, 2010; Gross et al., 2014). For instance, assessing the effectiveness of medical treatments (Singh & Mukhopadhyay, 2011), predicting equipment failure rates (de Cos Juez et al., 2010), or analyzing customer churn in the business domain (Danacica & Babucea, 2010). The primary goal of survival analysis is to estimate the probability (distribution) of an event occurring over time, given a set of covariates or risk factors. One of the key challenges in survival analysis involves dealing with censored data, which occurs when the event of interest is not observed for some individuals within the study period. This can happen due to various reasons, such as loss to follow-up, measurement failure, study termination, or the occurrence of competing risks (Salerno & Li, 2023). Handling censored data requires tailored statistical methods to avoid biased survival estimates. Another challenge is that

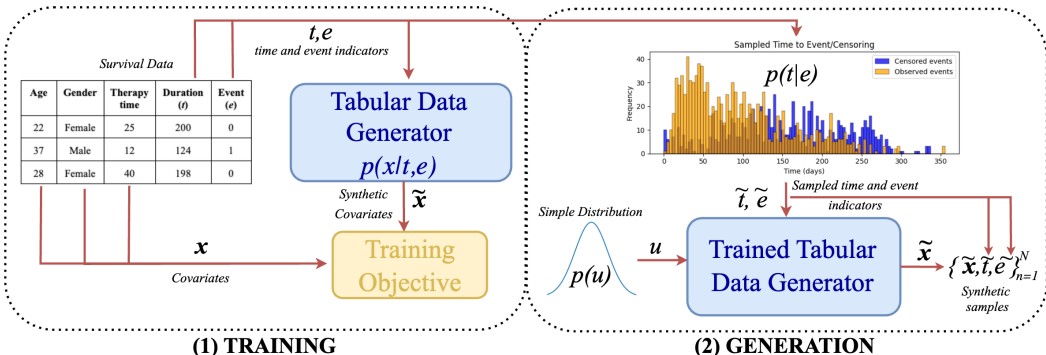

**(1) TRAINING**    **(2) GENERATION**

Figure 1: Block diagram of the proposed methodology. First, a conditional tabular data generator is trained to learn to sample covariates from $p(\boldsymbol{x}|t, e)$. After training, *both* event times $\tilde{t}$, and type $\tilde{e}$, are sampled from their joint distribution via $p(t|e)$ and $p(e)$ using their empirical distributions and passed into the trained generator along with $\boldsymbol{u} \sim p(\boldsymbol{u})$, where $p(\boldsymbol{u})$ is a simple distribution. The generator then repeatedly generates the synthetic covariates thus completing the synthetic dataset $\mathcal{D} = \{(\tilde{\boldsymbol{x}}, \tilde{t}, \tilde{e})\}_{n=1}^{N}$.

oftentimes, sample sizes in survival data are relatively small, or the proportion of observed events relative to those with censoring is small, thus causing overfitting issues which negatively impact generalization ability.

In most domains, such as clinical trials or engineering studies, collecting large amounts of survival data can be challenging, time-consuming, and costly. Synthetic data generation allows researchers to create large datasets with desired characteristics, enabling more robust model prototyping, development and evaluation. Synthetic survival data, which is predominantly tabular (or structured), can be generated using generative models that are specifically developed for tabular data, *e.g.*, autoencoders (Xu et al., 2019), adversarial generators (Yoon et al., 2020), diffusion generators (Kotelnikov et al., 2023), and even large language models (LLMs) (Borisov et al., 2022). However, apart from the well-known challenges associated with generating tabular data such as appropriately handling categorical and continuous data, mixed data types, as well as their joint distributions (Xu et al., 2019), survival data generation, especially in the medical domain, faces some unique challenges. These are due to mainly unavoidable differences in the distributions for observed and censored events, and their (unknown) underlying generation mechanism given the covariates. In practice, this challenge causes mismatches between these distributions when comparing real-world and synthetic data generated from it (Norcliffe et al., 2023). Consequently, such mismatches are likely to cause survival models trained on such synthetic data to underperform relative to the real-world data in terms of discrimination and calibration. So motivated, our work offers the following contributions:

- We propose a simple methodology for generating survival data by conditioning the generation of covariates on the event times and censoring indicators after sampling these from the empirical real-world data distributions as shown in Figure 1, thus $i$) readily resulting in matching observed and censoring distributions; and $ii$) allowing the user to choose from existing methods for conditional generation of tabular data without computational overhead.
- We show that our *generator-agnostic* methodology can be easily extended to use LLM-based tabular data generators for the generation of high-quality synthetic survival data, an application that to the best of our knowledge has not been explored so far.
- Extensive experiments on five real-world survival analysis datasets demonstrate the capabilities of the proposed methodology in terms of the quality of the generated observed, censored and covariate distributions, as well as the discrimination and calibration performance of survival analysis models trained on synthetic data and evaluated on real-world data. Moreover, we also show that our method offers better performance without compromising data privacy.

## 2 RELATED WORK

Generative models have emerged as powerful tools for synthesizing realistic data across various domains, including images, text, and tabular data. These models aim to learn the underlying probability distributions of the training data and generate new samples that exhibit similar characteristics.

Three prominent classes of generative models have gained significant traction: *generative adversarial networks* (GANs), *variational autoencoders* (VAEs), and *diffusion-based models*. GANs employ an adversarial training paradigm, where a generator network learns to produce synthetic data samples, while a discriminator network aims to distinguish between real and generated samples (Goodfellow et al., 2020). This adversarial process drives the generator to produce increasingly realistic samples. VAEs leverage variational inference techniques to learn a latent representation of the data, enabling the generation of new samples by sampling from the learned latent space (Kingma & Welling, 2013). Diffusion-based models, such as the *denoising diffusion probabilistic model* (DDPM) (Ho et al., 2020), gradually add noise to the data and then learn to reverse this process, generating new samples by denoising random points. These generative models have demonstrated remarkable success in various applications, including image synthesis (Kang et al., 2023), text generation (Su et al., 2022), and video generation (Jiang et al., 2023). In survival analysis, generative models have been applied to estimate event time distributions and hazard functions (Chapfuwa et al., 2018; Zhou et al., 2022).

Tabular data stands out as a prevalent data format in machine learning (ML), with more than 65% of datasets found in the Google Dataset Search platform[1] comprising tabular files, typically in comma-separated or spreadsheet formats (Benjelloun et al., 2020). While conventional generative methods are not optimally tailored for tabular data due to the mixture of continuous and categorical variables (Xu et al., 2019), modified versions have been developed for this domain. These include, the *conditional tabular generative adversarial network* (CTGAN) (Xu et al., 2019), which leverages the GAN framework to generate synthetic data preserving multivariate distributions and relationships, the *tabular variational autoencoder* (TVAE) (Xu et al., 2019), and the *anonymization through data synthesis using generative adversarial network* (ADS-GAN) (Yoon et al., 2020). The *tabular denoising diffusion probabilistic model* (TabDDPM) is a recent approach that leverages denoising diffusion probabilistic models to generate high-fidelity synthetic tabular data (Kotelnikov et al., 2023). Large language models (LLMs) have also shown potential for tabular data generation, using fine-tuning on token-represented tabular data (Borisov et al., 2022).

In synthetic survival data generation, early statistical models (Bender et al., 2005; Austin, 2012) transformed uniform samples into survival times but did not generate covariates. More recent techniques have incorporated deep learning into the generative process. Ranganath et al. (2016) proposed using deep exponential families to generate survival data, but this approach has limited flexibility on the learned distributions. Miscouridou et al. (2018) and Zhou et al. (2022) relaxed this assumption but still focused on generating survival times and censoring statuses conditioned on the covariates, rather than generating the covariates themselves. Recently, SurvivalGAN (Norcliffe et al., 2023) was developed, generating synthetic data in three steps: *i)* a conditional GAN (ADS-GAN) generates covariates ($\boldsymbol{x}$) and samples the event indicator ($e$) from the empirical distribution; *ii)* a survival function model (DeepHit (Lee et al., 2018)) predicts survival functions for the generated covariates; and *iii)* these outputs are used by a regression model (XGboost (Chen & Guestrin, 2016)) to predict the event time ($t$), generating the complete triplet ($\boldsymbol{x}, t, e$). While effective, this method is complex with multiple models, each having their own limitations. Alternatively, our work explores a much simpler method that adopts existing tabular data generators for survival data without the need for dedicated networks for the prediction of the survival function or event/censoring distributions.

## 3 METHODS

**Problem definition**  Instances (or subjects) from survival data can be represented in general as the triplet $z = (\boldsymbol{x}, t, e)$. Here, $\boldsymbol{x} \in \mathcal{X}$ denotes $m$-dimensional tabular covariates that describe an instance's state at an initial (or index) time, encompassing both continuous and categorical covariates. Then, $t_i \in \mathcal{T}$ represents the time of a specific event relative to the initial time, thus $t \geq 0$ and $\mathcal{T} \equiv \mathbb{R}_+$. Lastly, $e_i \in \mathcal{E}$ stands for the event indicator, commonly $\mathcal{E} = \{0, 1\}$, where $e = 1$ indicates the event of interest occurs at time $t$, whereas $e = 0$ signifies the event of interest has not occurred up to time $t$. In this work we only consider *right censoring* as it is the predominant form in real-world datasets, however, the proposed method can be readily extended to left or interval censoring.

**Background**  Survival analysis is a statistical framework used to analyze and model the time until the occurrence of the event of interest, also known as the survival time or time-to-event. Survival analysis

---

[1]https://datasetsearch.research.google.com/.

involves modeling the conditional probability density function $p(t|\boldsymbol{x})$, to estimate the likelihood of the event of interest occurring at time $t$ given the covariates $\boldsymbol{x}$. From this, the survival function is derived, representing the probability that the event has not taken place by time $t$, *i.e.*,

$$S(t \mid \boldsymbol{x}) = \int_t^\infty p\left(t' \mid \boldsymbol{x}\right) dt', \tag{1}$$

where $S(t \mid \boldsymbol{x})$ is an estimate of the proportion of instances (subjects) with covariates $\boldsymbol{x}$ who have survived up to time $t$. When the initial time is zero and given that events cannot occur at $t \leq 0$, thus $S(0|\boldsymbol{x}) = 1$. Additionally, since $p(t|\boldsymbol{x})$ is a valid probability distribution (non-negative), then $S(t|\boldsymbol{x})$ is a monotonically decreasing function. Time-to-event approximation involves estimating the expected lifetime for any given covariate value, denoted as $\mu(\boldsymbol{x})$. Specifically, this is obtained as $\mu(\boldsymbol{x}) = \int_0^\infty t' p(t'|\boldsymbol{x}) dt'$, which, through integration by parts, simplifies to the area under the survival curve: $\mu(\boldsymbol{x}) = \int_0^\infty S(t|\boldsymbol{x}) dt$. Survival models typically fall into one of two categories: $i$) parametric such as the accelerated failure time (Weibull, 1951), and log-logistic (Prentice, 1976) models; or $ii$) non-parametric such as the Kaplan-Meier estimator (Kaplan & Meier, 1958) and Cox proportional hazards model (Cox, 1972). Moreover, deep-learning versions of these have been proposed, *e.g.*, DeepSurv (Katzman et al., 2018), DeepHit (Lee et al., 2018), DATE (Chapfuwa et al., 2018), *etc*.

### 3.1 CONDITIONING ON EVENT TIME AND TYPE

Synthetic survival data generation involves the generation of samples from the complete joint distribution $p(\boldsymbol{x}, t, e)$. In practice, one can either sample from it directly (and unconditionally) using generative models for tabular data, or via conditioning using for instance $p(t|\boldsymbol{x}, e)p(\boldsymbol{x})p(e)$ or $p(\boldsymbol{x}|t, e)p(t|e)p(e)$. The former is the approach used in Norcliffe et al. (2023), which samples $\tilde{x}$ and $\tilde{e}$, from the marginals $p(\boldsymbol{x})$ and $p(e)$, are obtained using a conditional GAN (ADS-GAN) generator and the empirical distribution for the event indicators, respectively, and subsequently, samples $\tilde{t}$ from the conditional $p(t|\boldsymbol{x}, e)$ are generated (deterministically) using a regression model. One important drawback of this approach is that the quality of the samples for event times $\tilde{t}$ from $p(t|\boldsymbol{x}, e)$ is both dependent on the quality of the approximation $\tilde{t} \sim p_\phi(t|\boldsymbol{x}, e)$ (with parameters $\phi$) and that of $p(\boldsymbol{x})$ via $\tilde{\boldsymbol{x}} \sim p_\psi(\boldsymbol{x}|\boldsymbol{u})$ parameterized by $\psi$, and $\boldsymbol{u}$ being sampled from a simple distribution, *e.g.*, uniform or Gaussian. As a result, approximation error in covariates $\boldsymbol{x}$ compounds with that for $t$ resulting in event and censoring distributions that do not necessarily match the real data. Consequently, Norcliffe et al. (2023) also proposed metrics to quantify the quality of these distributions (see Section 4).

In an effort to alleviate these key issue, we reverse the conditioning and instead sample *both* event times and type from their joint distribution via $p(t|e)$ and $p(e)$, using their empirical distributions. Note that this is possible by assuming without loss of generality that the observed and censoring times are conditionally independent given the covariates, which also aligns with the common assumption of censoring at random in survival analysis, which posits that the censoring mechanism is independent of the unobserved survival times, conditional on the covariates. Then, we sample the covariates from $p(\boldsymbol{x}|t, e)$ using a conditional generator as follows

$$\tilde{e} \sim p(e), \qquad \tilde{t} \sim p(t|\tilde{e}), \qquad \boldsymbol{u} \sim p(\boldsymbol{u}), \qquad \tilde{\boldsymbol{x}} \sim p_\theta(\boldsymbol{x}|\tilde{t}, \tilde{e}, \boldsymbol{u}), \tag{2}$$

where $p_\theta(\boldsymbol{x}|\tilde{t}, \tilde{e}, \boldsymbol{u})$ is a conditional generator parameterized by $\theta$, while $p(\boldsymbol{u})$ is a simple distribution. Repeatedly sampling from the mechanism in equation 2 allows one to obtain a synthetic dataset $\mathcal{D} = \{(\boldsymbol{x}_n, t_n, e_n)\}_{n=1}^N$ whose empirical conditionals for event and censoring times readily match the ground-truth distributions, $p(t|e = 1)$ and $p(t|e = 0)$, respectively, and synthetic covariates that acknowledge their association with the event of interest while accounting for censoring. Importantly, using equation 2: $i$) eliminates the need for a separate model to generate event times (XGboost in Norcliffe et al. (2023)); $ii$) eliminates the need for a separate model to generate survival distributions (DeepHit in Norcliffe et al. (2023)), and $iii$) guarantees the quality of the observed and censored event distributions. Moreover, and from a practical perspective, equation 2 offers flexibility since $p_\theta(\boldsymbol{x}|\tilde{t}, \tilde{e}, \boldsymbol{u})$ can be modeled, in principle, with any conditional generator. In the experiments (see Section 4), we will consider TVAE, CTGAN, ADS-GAN, TabDDPM and LLMs. Note that in equation 2 we are not required to sample from the empirical distributions for $p(t)$ and $p(e)$, for instance one may alternatively fit univariate (kernel) density estimators and then draw $\tilde{t}$ and $\tilde{e}$ accordingly, especially, if the dataset is small and the number of unique values of $t$ in $\mathcal{D}$ is small.

## 3.2 Adapting Conditional Tabular Generators to Survival Data

Existing tabular generators (see Section 2) use distinct strategies to implement conditioning. Below we briefly describe how they are adapted to the considered survival data generation problem.

**CTGAN** This model being a conditional adversarial generator, synthesizes data using $G(\boldsymbol{u}, \boldsymbol{c})$, where $G(\cdot)$ is the generator specified as a neural network, $\boldsymbol{u}$ is a vector sampled from a simple distribution, *e.g.*, a standard Gaussian distribution, *e.g.*, $\boldsymbol{u} \sim \mathcal{N}(\boldsymbol{0}, \boldsymbol{I})$, and $\boldsymbol{c}$ is a one-hot vector representing a discrete conditioning covariate. See Xu et al. (2019) for additional details. In order to use $G(\boldsymbol{u}, \boldsymbol{c})$ as a sampling mechanism for $p_\theta(\boldsymbol{x}|\tilde{t}, \tilde{e}, \boldsymbol{u})$ in equation 2 we simply set $\boldsymbol{c} = E_t(\tilde{t}) \oplus \tilde{e}$, where $E_t(\cdot)$ is an $m$-dimensional sinusoidal time embedding (Wang & Chen, 2020) and $\oplus$ is the concatenation operator. In all our experiments we set $m = 4$.

**TVAE** The autoencoding formulation in Xu et al. (2019) does not specify explicitly how to perform conditional generation for the tabular VAE. However, the simplest strategy involves setting the encoder and decoder pair as $\boldsymbol{u} \sim \mathcal{N}(\mu(\boldsymbol{x}), \sigma^2(\boldsymbol{x}))$ and $\tilde{\boldsymbol{x}} \sim p_\theta(\boldsymbol{x}|\boldsymbol{c}, \boldsymbol{u})$, respectively, where here $\boldsymbol{u}$ is the latent representation for covariates $\boldsymbol{x}$, $\mu(\boldsymbol{x})$ and $\sigma^2(\boldsymbol{x})$ are two neural networks for the mean and variance functions of the latent representation $\boldsymbol{u}$, $p_\theta(\boldsymbol{x}|\boldsymbol{c}, \boldsymbol{u})$ is a probabilistic decoder specified using neural networks (see Xu et al. (2019) for details), $\boldsymbol{c}$ is a one-hot vector as above for CTGAN, and the input to the decoder conveniently implemented by concatenating $\boldsymbol{z}$ and $\boldsymbol{c}$. Similar to CTGAN, we make $\boldsymbol{c} = E_t(\tilde{t}) \oplus \tilde{e}$ in our implementation to sample from $p_\theta(\boldsymbol{x}|\tilde{t}, \tilde{e}, \boldsymbol{u})$ in equation 2 via $p_\theta(\boldsymbol{x}|\boldsymbol{c} = E_t(\tilde{t}) \oplus \tilde{e}, \boldsymbol{u})$.

**ADS-GAN** This alternative adversarial model specification encourages de-identifiability by letting the generator be $\tilde{\boldsymbol{x}} = G(\boldsymbol{u}, \boldsymbol{x}, \boldsymbol{c})$, *i.e.*, covariates $\boldsymbol{x}$ are also used as inputs to the generation function $G(\cdot)$, to encourage the model to generate samples $\tilde{\boldsymbol{x}}$ that are distinct from $\boldsymbol{x}$ to preserve privacy. See Yoon et al. (2020) for additional details. Consistent with CTGAN and TVAE above, we simply set $\boldsymbol{c} = E_t(\tilde{t}) \oplus \tilde{e}$.

**TabDDPM** This model designed specifically for tabular data employs a combination of Gaussian and multinomial diffusion processes to handle numerical and categorical features, respectively. Notably, each covariate uses a separate forward diffusion processes. The reverse diffusion function in Kotelnikov et al. (2023) is set as $\boldsymbol{x}_{is} = g_i(\boldsymbol{x}_i, \boldsymbol{x}_{i0}, s)$, where $g_i(\cdot)$ is modeled using neural networks with identity and softmax activations for continuous and discrete covariates, respectively, $\boldsymbol{x}_{is} = h_x(\boldsymbol{x_i}) + h_s(E_t(s)) + E_c(\boldsymbol{c})$ is the representation of the $i$-th covariate in $\boldsymbol{x}$ at diffusion step $s$, $h_x(x_i)$ is a fully connected layer with linear activation, $h_s(\cdot)$ is composed of two fully connected layers with sigmoid linear activations, $E_c(\cdot)$ is a standard (trainable) categorical embedding, and $s = 0, \ldots, S$, is such that $\boldsymbol{x}_{iS} \sim \mathcal{N}(\boldsymbol{0}, \boldsymbol{I})$ or $\boldsymbol{x}_{iS} \sim \text{Cat}(\boldsymbol{1}/K_i)$, for $K_i$ categories (distinct values), for continuous or discrete covariates, respectively. Note that effectively, $g_i(\cdot)$ models the residuals of $\boldsymbol{x}_{is}$ at diffusion step $s$ rather than $\boldsymbol{x}_{is}$ itself (Nichol & Dhariwal, 2021). For additional details of the formulation and and components of the model architecture see Kotelnikov et al. (2023). For our implementation, we set $\boldsymbol{c} = E_t(\tilde{t}) + E_s(\tilde{e})$ and set $m = 128$ as the embedding dimension.

## 4 Experiments

**Baselines and setup** We compare our methodology against the following baselines: generative adversarial networks for anonymization (ADS-GAN) (Yoon et al., 2020); conditional generative adversarial networks for tabular data (CTGAN) (Xu et al., 2019); variational autoencoder for tabular data (TVAE) (Xu et al., 2019); tabular denoising diffusion probabilistic models (TabDDPM) (Kotelnikov et al., 2023); and SurvivalGAN (Norcliffe et al., 2023). Note that only the latter is specific to survival data, whereas all the others generate tabular data *unconditionally*, *i.e.*, from the joint $p(\boldsymbol{x}, t, e)$. For CTGAN, TVAE, ADS-GAN, and TabDDPM models, we report metrics both directly using the models for survival data generation as well as our methodology, *i.e.*, using them as conditional generators given event times and censoring indicators sampled from the empirical distribution of the real data as described in Section 3.2. To evaluate downstream performance, survival models are trained on synthetic data and tested on real data using the Train on Synthetic Test on Real (TSTR) paradigm (Esteban et al., 2017). Specifically, the original dataset is divided into three folds, and the synthetic data generator is trained on two folds while the third is reserved for testing. Synthetic data equivalent (in

size) to the training data is then generated, and downstream models are trained on this synthetic dataset and evaluated on the held-out *real test set*. This process is repeated for all three fold combinations. We consider various survival models: linear (CoxPH) (Cox, 1972), gradient boosting (SurvivalXGBoost) (Barnwal et al., 2022), and neural networks (DeepHit) (Lee et al., 2018), and report metrics for the best-performing model. For each dataset, benchmark, and experimental setting, we report mean and standard deviation of performance metrics using 5 random seeds. To streamline the benchmarking, we utilized the Synthcity library (Qian et al., 2024), which provides implementations of a variety of synthetic tabular data generation models and benchmarking utilities. Detailed experimental settings and hyperparameters are in Appendix B.3. The source code for reproducing experiments is available at https://github.com/anonymous-785/synthetic_survival_data.

**Datasets**  We benchmark our methodology on a variety of real-world medical datasets. Specifically: *i*) *Study to understand prognoses preferences outcomes and risks of treatment* (SUPPORT) (Knaus et al., 1995); *ii*) *Molecular taxonomy of breast cancer international consortium* (METABRIC) (Curtis et al., 2012); *iii*) *ACTG 320 clinical trial dataset* (AIDS) (Hammer et al., 1997); *iv*) *Rotterdam & German breast cancer study group* (GBSG) (Schumacher et al., 1994); and *v*) *Assay of serum free light chain* (FLCHAIN) (Dispenzieri et al., 2012). See Appendix B.2 for additional details.

**Metrics**  To evaluate the quality of the generated synthetic survival data, various metrics are employed, which can be categorized into three groups: *covariates quality*, *event-time distribution quality*, and *downstream performance*. For assessing the quality of the generated covariates $\tilde{x}$, the Jensen-Shannon (JS) distance and Wasserstein distance (WS) are used to measure the divergence between the generated and original covariate distributions. We also measure the differences between the covariates in an univariate fashion using hypothesis testing, namely, Wilcoxon rank-sum and Chi squared tests for continuous and discrete covariates respectively, and then summarize the obtained $p$-values for all covariates as the proportion (PVP) below the standard significance threshold $\alpha = 0.05$ after correction for multiple testing via Benjamini-Hochberg (Benjamini & Hochberg, 1995). For the quality of the event time distributions we quantify the alignment between ground-truth and generated temporal marginals, namely, $p(t, e)$ is evaluated using the Kaplan-Meier (KM) divergence, optimism, and short-sightedness metrics as previously described in (Norcliffe et al., 2023). The KM divergence compares the mean absolute difference between the synthetic and real survival function estimates, while optimism and short-sightedness are a proxy for their bias and variance, respectively. These three metrics capture the accuracy of the generated censoring and event distributions. Finally, to assess downstream performance, survival models are trained on the synthetic data and evaluated on real dataset. Specifically, we consider the concordance index (C-index) (Harrell et al., 1982) and the Brier score (Brier, 1950). The former measures the discriminative ability of the survival model, while the latter quantifies the calibration of the probabilistic predictions.

### 4.1 Synthetic Survival Data Generation Benchmark

**Covariate quality metrics:** Results in Table 1 compare the similarity between the distribution of synthetic samples and the original data. First, we assess the overall (covariance) structure of the synthetic covariates relative to the original data via the JS and WS distances. Then, we perform hypothesis testing to compare the (univariate) marginal distributions of each covariate relative to the original data. Specifically, we use Wilcoxon rank-sum and Chi squared tests for continuous and discrete covariates, respectively, as described above. Importantly, since we sample $\tilde{t}$ and $\tilde{E}$ directly from the empirical (training) distributions it is clear that the synthetic and original distributions for event times accounting for censoring match, thus we do not report KM divergence, optimism and short-sightedness in Table 1, however, they are reported in Appendix C for completeness. Our models outperformed or matched baselines in all 5 datasets for JS distance, and surpassed them in all 5 for WS distance. For PVP, we outperformed baselines in 3 of 5 datasets. This aligns with expectations, as modeling mixed-data types remains challenging in tabular data generation (Xu et al., 2019). The PVP metric reveals our method's performance is bounded by current conditional generator capabilities. Notably, in Figure 2, we directly compare the distribution of $p$-values for the best-performing conditional model with that of the best unconditional model for a given dataset using quantile-quantile (Q-Q) plots. We observe that our methodology leads to better $p$-value distributions, *i.e.*, our synthetic datasets are more consistent with the null (uniform) $p$-value distribution. In the

Table 1: Quality (JS Distance, WS Distance, and PVP) and downstream (C-Index and Brier Score) metrics. Models conditioning on $t$ and $e$ are highlighted † (our method), UM refers to the best-performing unconditional model among TVAE, TabDDPM, CTGAN and ADS-GAN, and Original is for the survival model trained on the real (training) data. Error bars are standard deviations for 5 repetitions.

| Metric | Method | AIDS | METABRIC | SUPPORT | GBSG | FLCHAIN |
|---|---|---|---|---|---|---|
| JS distance (↓) | SurvivalGAN | 0.013±0.00 | 0.009±0.00 | 0.008±0.00 | 0.008±0.00 | 0.009±0.00 |
| | TVAE† | 0.007±0.00 | 0.008±0.00 | **0.004±0.00** | 0.005±0.00 | 0.002±0.00 |
| | TabDDPM† | 0.007±0.00 | **0.007±0.00** | 0.013±0.00 | 0.005±0.00 | **0.001±0.00** |
| | CTGAN† | 0.013±0.00 | 0.020±0.01 | 0.005±0.00 | **0.003±0.00** | 0.004±0.00 |
| | ADS-GAN† | **0.006±0.00** | 0.009±0.00 | 0.005±0.00 | 0.004±0.00 | 0.010±0.01 |
| | UM | **0.006±0.00** | **0.007±0.00** | 0.005±0.00 | 0.005±0.00 | 0.002±0.00 |
| WS distance (↓) | SurvivalGAN | 0.112±0.01 | 0.039±0.00 | 0.043±0.00 | 0.019±0.00 | 0.052±0.00 |
| | TVAE† | **0.061±0.00** | **0.028±0.00** | **0.032±0.00** | 0.013±0.00 | **0.016±0.00** |
| | TabDDPM† | 0.159±0.02 | 0.089±0.00 | 0.308±0.02 | 0.056±0.00 | 0.028±0.00 |
| | CTGAN† | 0.095±0.00 | 0.133±0.01 | 0.034±0.00 | 0.013±0.00 | 0.019±0.00 |
| | ADS-GAN† | 0.082±0.00 | 0.037±0.00 | 0.036±0.00 | **0.011±0.00** | 0.018±0.00 |
| | UM | 0.069±0.00 | 0.031±0.00 | 0.036±0.00 | 0.013±0.00 | 0.016±0.00 |
| PVP (↓) | SurvivalGAN | 0.181±0.00 | 0.555±0.00 | 0.571±0.00 | 0.485±0.00 | 0.555±0.00 |
| | TVAE† | **0.090±0.00** | 0.444±0.00 | 0.457±0.06 | **0.142±0.00** | **0.222±0.04** |
| | TabDDPM† | 0.181±0.06 | 0.222±0.00 | 0.528±0.03 | 0.199±0.07 | **0.222±0.04** |
| | CTGAN† | 0.272±0.00 | 0.555±0.00 | 0.428±0.00 | 0.571±0.00 | 0.511±0.06 |
| | ADS-GAN† | 0.309±0.04 | 0.555±0.00 | 0.600±0.03 | 0.428±0.00 | 0.422±0.04 |
| | UM | 0.096±0.04 | **0.000±0.00** | **0.171±0.08** | 0.200±0.00 | 0.244±0.00 |
| C-Index (↑) | SurvivalGAN | 0.735±0.00 | 0.625±0.00 | 0.602±0.00 | 0.668±0.00 | 0.870±0.00 |
| | TVAE† | 0.737±0.00 | 0.612±0.00 | 0.583±0.00 | 0.672±0.00 | 0.872±0.00 |
| | TabDDPM† | 0.660±0.07 | 0.589±0.01 | 0.536±0.00 | 0.663±0.00 | 0.876±0.00 |
| | CTGAN† | 0.746±0.00 | 0.628±0.01 | 0.577±0.00 | 0.665±0.01 | 0.874±0.00 |
| | ADS-GAN† | **0.797±0.01** | **0.655±0.00** | 0.623±0.00 | **0.684±0.00** | **0.880±0.00** |
| | UM | 0.779±0.00 | 0.649±0.00 | **0.625±0.00** | 0.679±0.00 | 0.879±0.00 |
| | Original | 0.760±0.00 | 0.636±0.00 | 0.616±0.00 | 0.695±0.00 | 0.870±0.00 |
| Brier Score (↓) | SurvivalGAN | 0.068±0.00 | 0.205±0.00 | 0.202±0.00 | 0.212±0.00 | 0.096±0.00 |
| | TVAE† | **0.059±0.00** | 0.199±0.00 | 0.207±0.00 | 0.214±0.00 | 0.095±0.00 |
| | TabDDPM† | 0.063±0.00 | 0.212±0.00 | 0.217±0.00 | 0.215±0.00 | 0.096±0.00 |
| | CTGAN† | 0.061±0.00 | 0.199±0.00 | 0.205±0.00 | 0.215±0.01 | 0.089±0.00 |
| | ADS-GAN† | **0.059±0.00** | **0.197±0.00** | **0.198±0.00** | 0.213±0.00 | **0.084±0.00** |
| | UM | 0.060±0.00 | 0.200±0.00 | 0.199±0.00 | **0.207±0.00** | 0.086±0.00 |
| | Original | 0.062±0.00 | 0.200±0.00 | 0.195±0.00 | 0.205±0.00 | 0.095±0.00 |

case where our methodology underperforms shown in Figure 2b, the performance is not substantially worse than the baseline (unconditional) TabDDPM model. Full results are shown in Appendix C.

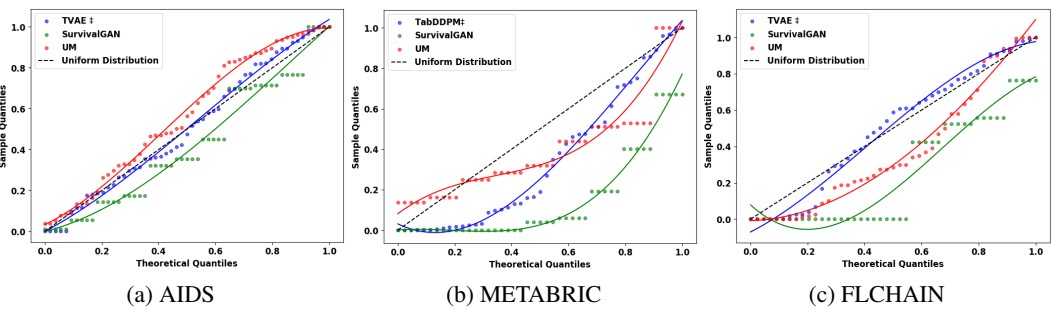

| (a) AIDS | (b) METABRIC | (c) FLCHAIN |
|---|---|---|

Figure 2: Q-Q plots comparing the $p$-value distributions of the best-performing conditional model († highlights our method) with that of the best unconditional model (UM). The dashed line represents the expected (uniform) distribution.

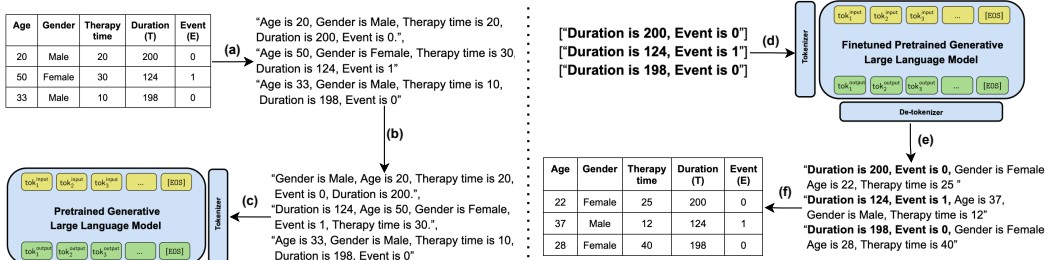

Figure 3: Training and sampling procedure for survival data generation using LLMs.

**Downstream Performance** We conduct a comparative analysis of survival models trained with synthetic data generated by our methodology against models trained with data from baseline methods. A favorable outcome is achieved when a model trained with synthetic data performs comparably to or occasionally even better than a model trained with real data, while also outperforming models trained with alternative synthetic data sources. For reference, we also report the C-Index and Brier Score for survival models trained on the original data. C-index and Brier score serve as the most widely used indicators of performance, as they encapsulate the entire conditional distribution of covariates, event/censoring times, and event indicators $p(t, e|x)$. Results in Table 1 demonstrate that in both of these metrics, we outperform the baselines in 4 of 5 datasets. Further, in most cases, we were also able to achieve better performance than survival models trained on the original data.

### 4.2 Fine Tuning an LLM for Survival Data Generation

Generation of realistic tabular data (GReaT) is a recently proposed approach to generating high-quality synthetic tabular data using LLMs (Borisov et al., 2022). This is achieved by representing the tabular data as a sequence of text and training the language model to generate new sequences that correspond to valid and plausible tabular data instances. We adapt GReaT to generate synthetic survival data by conditioning the generation on time-to-event and event-type. The fine-tuning of a pre-trained auto-regressive LLM on the encoded tabular data for data generation as proposed in Borisov et al. (2022) involves the following steps. *Textual encoding and feature permutation:* The tabular data with $M$ column names $\{f_m\}_{m=1}^M$ and thus, $M$-dimensional rows $\{x_n\}_{n=1}^N$ are converted into textual representation. Each row (sample) $x_n$ is encoded as a sentence with elements $t_n = \{t_{nm}\}_{m=1}^M$, where $t_{nm} = [f_m, "is", x_{nm}, ", "]$ contains the column name $f_m$ and its value $x_{nm}$. *Model training:* The LLM is trained using DistilGPT2 (Li et al., 2021) on the textually encoded dataset $\{t_n\}_{n=1}^N$, with elements of $t_n$ permuted at random to remove pseudo-positional information as column order in a tabular dataset is in principle non-informative. *Sampling:* Feature permutations during training enable the model to start generation with any combination of features and values. To generate synthetic data conditionally, we prompt the trained model with conditioning sequences sampled from the empirical marginal $p(t, e)$, and let it generate the remaining tokens to complete the textual feature vector, which is then converted back to tabular format. Unconditional generation follows Borisov et al. (2022). The training and sampling procedure is shown in Figure 3. Table 2 compares the performance of GReaT with and without conditional generation, against the best generator from Table 1 (results shown for two datasets). See Appendix C for full results including Q-Q plots. We observe that conditional generation consistently enhances GReaT's performance over the unconditional variant and baseline generators. Further, PVP also improves significantly, outperforming all unconditional models across all datasets, underscoring the effectiveness of the LLM in modeling univariate marginals. Note however that GReaT is much more costly compared to other models as shown in Appendix B.1.

### 4.3 Sub-population Level Evaluation of Synthetic Data

In this experiment, we evaluate the performance of the proposed methodology at the sub-population level using the AIDS dataset, using race (White, Black and Hispanic) to define the sub-populations. Performance evaluation is carried out via race-stratified $K$-fold cross-validation. We consider survival models in three scenarios: $i)$ trained on the real data; $ii)$ trained on synthetic data with the same race proportion as the original data (*Synthetic*); and $iii)$ trained on synthetic data with balanced race samples while preserving the distribution of observed and censored events for each race (*Synthetic*

Table 2: Quality (JS, WS distance and PVP) and downstream (C-Index and Brier Score) metrics. Models conditioning on $t$ and $e$ are highlighted † (our method). BM refers to the best-performing model from Table 1.

| Dataset | Method | C-Index | Brier Score | JS distance | WS Distance | PVP |
|---|---|---|---|---|---|---|
| **AIDS** | SurvivalGAN | 0.735±0.00 | 0.068±0.01 | 0.013±0.00 | 0.12±0.00 | 0.181±0.00 |
| | GReaT† | 0.790±0.00 | 0.063±0.00 | **0.003±0.00** | **0.036±0.00** | **0.000±0.00** |
| | GReaT | 0.725±0.01 | 0.063±0.00 | 0.004±0.00 | 0.046±0.00 | 0.090±0.00 |
| | BM | **0.797±0.01** | **0.059±0.00** | 0.006±0.00 | 0.061±0.00 | 0.090±0.00 |
| **FLCHAIN** | SurvivalGAN | 0.870±0.00 | 0.096±0.00 | 0.009±0.00 | 0.052±0.00 | 0.555±0.00 |
| | GReaT† | **0.880±0.00** | **0.082±0.00** | **0.001±0.00** | **0.015±0.00** | 0.111±0.00 |
| | GReaT | 0.878±0.00 | 0.090±0.00 | **0.001±0.00** | 0.020±0.00 | 0.222±0.00 |
| | BM | **0.880±0.00** | 0.084±0.00 | **0.001±0.00** | 0.016±0.00 | 0.222±0.04 |

Table 3: Downstream (C-Index and Brier Score) performance metrics for survival models trained on Real Data, *Synthetic*, and *Synthetic (Balanced)*. Models conditioning on $t$ and $e$ are highlighted † (our method).

| Method | Race | Synthetic | | Synthetic (Balanced) | |
|---|---|---|---|---|---|
| | | **C-index** | **Brier Score** | **C-index** | **Brier Score** |
| ADS-GAN† | All | 0.722±0.01 | 0.071±0.00 | 0.745±0.02 | 0.065±0.01 |
| | Race 1 | 0.722±0.00 | 0.066±0.00 | 0.729±0.00 | 0.062±0.00 |
| | Race 2 | 0.722±0.00 | 0.070±0.00 | 0.729±0.00 | 0.063±0.00 |
| | Race 3 | 0.763±0.01 | 0.070±0.00 | 0.758±0.01 | 0.063±0.00 |
| SurvivalGAN | All | 0.663±0.00 | 0.100±0.02 | 0.683±0.01 | 0.076±0.01 |
| | Race 1 | 0.663±0.00 | 0.092±0.00 | 0.676±0.00 | 0.072±0.01 |
| | Race 2 | 0.663±0.00 | 0.095±0.01 | 0.676±0.01 | 0.073±0.02 |
| | Race 3 | 0.668±0.01 | 0.095±0.01 | 0.698±0.01 | 0.073±0.01 |

| Method | Race | Real Data | |
|---|---|---|---|
| | | **C-index** | **Brier Score** |
| Original | All | 0.735±0.01 | 0.075±0.01 |
| | Race 1 | 0.724±0.00 | 0.069±0.00 |
| | Race 2 | 0.724±0.00 | 0.072±0.00 |
| | Race 3 | 0.778±0.02 | 0.072±0.00 |

*(Balanced)*). For the survival models trained on the original AIDS dataset, the C-index differs across races, with the model performing better on Hispanic (0.778) when compared to White (0.724) and Black (0.724), with a $0.778/0.724 \approx 1.07$ ratio. When training using our synthetic data (ADS-GAN conditioned on time and event) with the same distribution as the original data, the C-index values reflect a similar performance ratio of 1.06 between races. For the balanced distribution scenario, all performance metrics improve at the expense of reducing the performance ratio between Hispanic and White/Black observed in the original data to 1.04. Further, the proposed model consistently outperforms SurvivalGAN, which is less able to capture the race performance difference with ratios 1.01 and 1.03 for Synthetic and Synthetic (Balanced), respectively.

## 4.4 EVALUATING THE EFFECTS OF SAMPLING $t$ AND $e$ FROM A PRIVACY PERSPECTIVE

To explore the acceptability of bootstrapping $t$ and $e$ when generating synthetic data, we employed the Distance to Closest Record (DCR) metric to evaluate the privacy preservation capabilities of various synthetic data generation methods (Zhao et al., 2021). The DCR quantifies the Euclidean distance between each synthetic record and its nearest real counterpart. A higher DCR value indicates a lower risk of privacy breach. We report the median and minimum DCR for all synthetic survival data generators used in our study, with the addition of a Synthetic Minority Oversampling Technique (SMOTE) (Chawla et al., 2002) baseline. SMOTE, originally proposed for minority class oversampling, is a simple interpolation-based method that generates synthetic points as convex combinations of real data points and their $k$-th nearest neighbors. In this study, we generalized and applied SMOTE to synthetic data generation to bootstrap entire data point $(\tilde{x}, \tilde{t}, \tilde{e})$, for comparison purposes. The results, presented in Table 4, demonstrate that the median DCR for the methods where

Table 4: Median value of Distance of closest record (DCR) from the original. Models conditioning on $t$ and $e$ are highlighted † (our method). UM refers to the best-performing unconditional model among TVAE, TabDDPM, CTGAN and ADS-GAN. Error bars are standard deviations for 5 repetitions. The best (highest) values are in bold while the worst (lowest) values are underlined.

| Metric | Method | AIDS | METABRIC | SUPPORT | GBSG | FLCHAIN |
|---|---|---|---|---|---|---|
| Median DCR | SurvivalGAN | 1.035±0.00 | 0.969±0.00 | 1.589±0.00 | 0.500±0.00 | **0.796±0.00** |
| | TVAE† | 0.883±0.00 | 0.877±0.00 | 1.511±0.00 | 0.476±0.00 | 0.642±0.00 |
| | TabDDPM† | **1.172±0.03** | 0.908±0.01 | 1.612±0.03 | 0.519±0.00 | 0.572±0.01 |
| | CTGAN† | 0.918±0.01 | 1.043±0.00 | 1.594±0.00 | **0.524±0.00** | 0.695±0.00 |
| | ADS-GAN† | 1.133±0.17 | 0.992±0.00 | **1.691±0.00** | 0.519±0.00 | 0.667±0.01 |
| | SMOTE | 0.388±0.00 | 0.698±0.00 | 0.958±0.00 | 0.290±0.00 | 0.381±0.00 |
| | UM | 1.158±0.01 | **1.087±0.00** | 1.666±0.00 | 0.515±0.00 | 0.641±0.00 |

Table 5: Minimum value of Distance of closest record (DCR) from the original. Models conditioning on $t$ and $e$ are highlighted † (our method). UM refers to the best-performing unconditional model among TVAE, TabDDPM, CTGAN and ADS-GAN. Error bars are standard deviations for 5 repetitions. The best (highest) values are in bold while the worst (lowest) values are underlined.

| Metric | Method | AIDS | METABRIC | SUPPORT | GBSG | FLCHAIN |
|---|---|---|---|---|---|---|
| Minimum DCR | SurvivalGAN | 0.048±0.00 | 0.172±0.00 | 0.326±0.00 | 0.062±0.00 | 0.057±0.00 |
| | TVAE† | 0.077±0.03 | 0.202±0.02 | 0.370±0.02 | 0.033±0.00 | 0.026±0.00 |
| | TabDDPM† | 0.095±0.00 | 0.193±0.05 | 0.403±0.01 | **0.065±0.00** | 0.037±0.00 |
| | CTGAN† | **0.139±0.01** | **0.215±0.01** | 0.321±0.01 | 0.045±0.01 | 0.054±0.01 |
| | ADS-GAN† | 0.102±0.01 | 0.185±0.04 | 0.391±0.01 | 0.053±0.01 | **0.066±0.02** |
| | SMOTE | 0.000±0.00 | 0.000±0.00 | 0.000±0.00 | 0.000±0.00 | 0.000±0.00 |
| | UM | 0.109±0.02 | 0.213±0.00 | **0.429±0.00** | 0.050±0.00 | 0.055±0.00 |

$t$ and $e$ were bootstrapped (denoted by a dagger †) was higher in 3 of 5 data sets, although by a small margin. A similar observation can be made in Table 5 where minimum DCR was higher for our methods in 4 of 5 datasets. In general, the median and minimum DCR values were largely similar between the methods with and without conditioning on $t$ and $e$, suggesting that sampling them is not likely to impact privacy. However, SMOTE consistently exhibited the lowest DCR in all datasets, indicating potential privacy concerns. These findings provide empirical evidence that bootstrapping $t$ and $e$ is generally acceptable from a privacy perspective. However, we note that even the most stringent minimum DCR does not provide privacy guarantees, so it needs to be interpreted with care. Full results are shown in Appendix C.

## 5 CONCLUSION

This work proposed a simple yet effective methodology for generating high-quality synthetic survival data by conditioning the generation of covariates on event times and censoring indicators sampled from the empirical distributions. Through extensive experiments on multiple real-world datasets, we demonstrated that our approach outperforms several competitive baselines across various evaluation metrics that assess the quality of the generated covariate distributions, alignment with the ground-truth event time distributions, and the downstream performance of survival models trained on the synthetic data. Moreover, we showcased the applicability of LLMs for survival data generation by fine-tuning them in a conditional manner on the textual representations of tabular data and how the proposed method preserves the sub-population-level performance characteristics of real-world data while preserving patient privacy.

**Limitations** Despite its promising results, our work has limitations. First, the quality of the generated data is highly dependent on the representativeness and diversity of the original dataset used for training the generative models. If the training data exhibit biases or lack sufficient variability, these likely will propagate to the synthetic data. Second, while our approach ensures accurate reproduction of the event time and censoring distributions, it does not explicitly consider time-varying covariates, which may be relevant in certain applications. Finally, further research is needed to address bias and equity in survival data. Though we attempt to understand the behavior of survival models trained on synthetic data at a sub-population level, we acknowledge that bias and equity are multifaceted challenges extending beyond the scope of this study. These are exciting avenues for further research.

## 6 ETHICS STATEMENT

This study focuses on synthetic survival data generation, which has important ethical implications. While our method aims to preserve patient privacy by generating synthetic data, we acknowledge the potential risks of reinforcing biases present in the original datasets. We have made efforts to evaluate our approach across different sub-populations to assess fairness, but further work is needed to fully address bias and equity concerns in survival analysis. The synthetic data generated should not be used for real clinical decision-making without extensive validation. We have no conflicts of interest to declare. All datasets used are publicly available and de-identified. Our work complies with relevant data protection regulations. We encourage users of our method to carefully consider the ethical implications and potential biases when applying it to sensitive healthcare data.

## 7 REPRODUCIBILITY

To ensure reproducibility, we have provided detailed descriptions of our methodology, datasets, and experimental setup throughout the paper and appendix. The hyperparameters for all models are specified in Appendix B.3. We have made our code publicly available at the anonymous GitHub repository linked in Section 4. This includes implementations of our proposed method and baselines. The datasets used are all publicly available, with download links provided in Appendix B.2. We report results as means and standard deviations over multiple random seeds. Our computational resources and runtimes are described in Appendix B.1. By providing these details, we aim to enable other researchers to reproduce our experiments and build upon our work.

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

Table 6: Training and generation time for synthetic survival data generation (in seconds). Models conditioning on $t$ and $e$ are highlighted (†).

| Metric | Method | AIDS | METABRIC | SUPPORT | GBSG | FLCHAIN |
|---|---|---|---|---|---|---|
| TTPI (↓) | SurvivalGAN | 0.178±0.00 | 0.260±0.00 | 1.234±0.01 | 0.283±0.00 | 1.024±0.01 |
| | TVAE† | 0.079±0.00 | 0.129±0.00 | 0.717±0.00 | 0.137±0.00 | 0.520±0.00 |
| | TabDDPM† | 0.055±0.00 | **0.049±0.00** | **0.208±0.00** | **0.066±0.00** | **0.182±0.00** |
| | CTGAN† | 0.165±0.00 | 0.240±0.00 | 1.209±0.01 | 0.246±0.00 | 0.894±0.01 |
| | ADS-GAN† | 0.143±0.00 | 0.239±0.00 | 1.148±0.01 | 0.256±0.00 | 0.825±0.01 |
| | TVAE | 0.136±0.00 | 0.186±0.00 | 1.023±0.01 | 0.187±0.00 | 0.735±0.00 |
| | TabDDPM | **0.046±0.00** | 0.050±0.00 | 0.215±0.00 | 0.070±0.00 | 0.183±0.00 |
| | CTGAN | 0.193±0.00 | 0.282±0.00 | 1.312±0.01 | 0.287±0.00 | 1.028±0.01 |
| | ADS-GAN | 0.214±0.00 | 0.291±0.00 | 1.404±0.01 | 0.306±0.00 | 1.061±0.01 |
| GT (↓) | SurvivalGAN | 0.396±0.01 | 0.421±0.02 | 0.896±0.08 | 0.407±0.05 | 0.715±0.05 |
| | TVAE† | 0.089±0.00 | 0.105±0.00 | 0.376±0.05 | 0.119±0.02 | 0.251±0.01 |
| | TabDDPM† | 11.875±0.16 | 9.477±0.35 | 48.985±0.37 | 17.451±0.22 | 37.216±0.38 |
| | CTGAN† | **0.075±0.01** | 0.088±0.01 | 0.152±0.00 | 0.066±0.01 | 0.102±0.01 |
| | ADS-GAN† | **0.075±0.01** | **0.084±0.01** | **0.148±0.00** | **0.065±0.00** | **0.102±0.01** |
| | TVAE | 0.128±0.02 | 0.135±0.00 | 0.468±0.09 | 0.124±0.00 | 0.281±0.11 |
| | TabDDPM | 11.785±0.36 | 9.466±0.33 | 50.017±0.81 | 18.085±0.56 | 34.937±0.85 |
| | CTGAN | 0.079±0.00 | 0.087±0.00 | 0.192±0.03 | 0.073±0.00 | 0.124±0.11 |
| | ADS-GAN | 0.089±0.00 | 0.098±0.01 | 0.212±0.04 | 0.085±0.01 | 0.111±0.00 |

Table 7: Training and generation time for synthetic survival data generation using LLMs (in seconds). Models conditioning on $t$ and $e$ are highlighted (†).

| Metric | Method | AIDS | METABRIC | SUPPORT | GBSG | FLCHAIN |
|---|---|---|---|---|---|---|
| TTPI | GReaT† | 5.154±0.11 | 9.600±0.19 | 49.800±0.00 | 6.660±0.21 | 23.400±0.10 |
| GT | GReaT | 14.237±0.15 | 121.451±0.20 | 270.798±0.99 | 23.516±0.05 | 77.154±0.18 |
| | GReaT† | 623.156±2.00 | 912.126±1.76 | 5520±5.57 | 812.366±0.25 | 1140.520±2.59 |

## A  BROADER IMPACT

The ability to generate realistic synthetic survival datasets can have far-reaching impacts across various domains, especially in privacy-sensitive applications like healthcare and clinical research. Synthetic data can enable model development, benchmarking, and collaboration while preserving patient confidentiality and complying with data protection regulations. Furthermore, our methodology can potentially address the common challenge of limited data availability in survival analysis by augmenting existing datasets or creating entirely new synthetic datasets tailored to specific requirements. While synthetic survival data is specific to the domain to which it is applied, limiting the potential for misuse, it is important to acknowledge the possibility of reinforcing biases present in the training data, as is the case with any generative model. Though we aim to understand the behavior of survival models trained on synthetic data across sub-populations, we recognize that addressing bias and ensuring equity are complex challenges that extend beyond the scope of this study. Thus, it is crucial to exercise caution and implement appropriate safeguards to mitigate potential biases and promote fairness in the development and deployment of such models.

## B  EXPERIMENTAL DETAILS

### B.1  COMPUTATIONAL COST

All experiments, except for the LLM fine-tuning (see Section 4), were conducted on Google Colab Pro using a T4 GPU. For the LLM fine-tuning experiments, an NVIDIA A100 GPU was utilized on Colab. In Table 6 we report the training time per iteration (TTPI) along with the time taken for synthetic data generation (GT) for all models used in Section 4.1, while the training and generation time for Section 4.2) are reported in Table 7.

Table 8: Summary statistics of the datasets used in the study.

| Dataset | No. instances | No. censored instances | No. features |
|---------|---------------|------------------------|--------------|
| AIDS | 1151 | 96 | 11 |
| METABRIC | 1904 | 801 | 9 |
| FLCHAIN | 7874 | 5705 | 9 |
| GBSG | 2232 | 965 | 7 |
| SUPPORT | 8873 | 2837 | 14 |

## B.2 DATASETS

We benchmark our methodology on a variety of medical datasets summarized in Table 8. Specifically: $i$) Study to understand prognoses preferences outcomes and risks of treatment (SUPPORT) (Knaus et al., 1995); $ii$) Molecular taxonomy of breast cancer international consortium (METABRIC) (Curtis et al., 2012); $iii$) ACTG 320 clinical trial dataset (AIDS) (Hammer et al., 1997); $iv$) Rotterdam & German breast cancer study group (GBSG) (Schumacher et al., 1994); and $v$) Assay of serum free light chain (FLCHAIN) (Dispenzieri et al., 2012). Pre-processed versions of METABRIC, SUPPORT, and GBSG can be found at: `https://github.com/havakv/pycox`. AIDS and FLCHAIN datasets can be downloaded from `https://github.com/sebp/scikit-survival/tree/master/sksurv/datasets/data`. For the FLCHAIN dataset, missing values in continuous covariates were imputed to the mean, while in discrete covariates they were imputed to the mode. All of these datasets are publicly available hence the experiments can be readily reproduced. In parts of our code (see Section 3.2 and 4), we utilize and modify the Synthcity library (`https://github.com/vanderschaarlab/synthcity`) which is protected under the *Apache-2.0* license. All rights to Synthcity are reserved by the original authors (Qian et al., 2024).

## B.3  Hyperparameters

For reproducibility purposes, all hyperparameters are specified below. Table 9 lists the hyperparameters for the downstream survival models used in the benchmarks. Further, Tables 10 and 11 provide the hyperparameters for all generative models employed in the study.

Table 9: Hyperparameters for the survival models used in Section 4.

| Method | Parameter | Parameter Value |
|---|---|---|
| CoxPH | Estimation Method | Breslow |
| | Penalizer | 0.0 |
| | $L^1$ Ratio | 0.0 |
| SurvivalXGBoost | Objective | Survival: AFT |
| | Evaluation Metric | AFT Negative Log Likelihood |
| | AFT Loss Distribution | Normal |
| | AFT Loss Distribution Scale | 1.0 |
| | No. Estimators | 100 |
| | Column Subsample Ratio (by node) | 0.5 |
| | Maximum Depth | 5 |
| | Subsample Ratio | 0.5 |
| | Learning Rate | $5 \times 10^{-2}$ |
| | Minimum Child Weight | 50 |
| | Tree Method | Histogram |
| | Booster | Dart |
| Deephit | No. Durations: | 1000 |
| | Batch Size | 100 |
| | Epochs | 2000 |
| | Learning Rate | $1 \times 10^{-2}$ |
| | Hidden Width | 300 |
| | $\alpha$ | 0.28 |
| | $\sigma$ | 0.38 |
| | Dropout Rate | 0.2 |
| | Patience | 20 |

Table 10: Hyperparameters used for the LLM in Section 4.2.

| Method | Parameter | Parameter Value |
|---|---|---|
| GReaT (DistilGPT2) | Batch Size | 32 |
| | No. Iterations | 1000 |
| | Learning Rate | $5 \times 10^{-5}$ |
| | Optimizer | AdamW |
| | Sampling Temperature | 0.7 |
| | Sampling Batch Size | 100 |

Table 11: Hyperparameters of the generative models used in synthetic benchmarks in Section 4.1.

| Model | Parameter | Parameter Value |
|---|---|---|
| ADS-GAN | No. Iterations | 10000 |
| | Generator no. Hidden Layers | 2 |
| | Generator Hidden Units | 500 |
| | Generator Non-linearity | ReLU |
| | Generator Dropout Rate | 0.1 |
| | Discriminator No. Hidden Layers | 2 |
| | Discriminator Hidden Units | 500 |
| | Discriminator Non-linearity | Leaky ReLU |
| | Discriminator Dropout Rate | 0.1 |
| | Learning Rate | $1 \times 10^{-3}$ |
| | Weight Decay | $1 \times 10^{-3}$ |
| | Batch Size | 200 |
| | Gradient Penalty ($\lambda$) | 10 |
| | Identifiability Penalty | 0.1 |
| | Encoder Max Clusters | 5 |
| | Early Stopping Patience | 5 |
| CTGAN | No. Iterations | 2000 |
| | Generator No. Hidden Layers | 2 |
| | Generator Hidden Units | 500 |
| | Generator Non-linearity | ReLU |
| | Learning Rate | $1 \times 10^{-3}$ |
| | Weight Decay | $1 \times 10^{-3}$ |
| | Discriminator No. Hidden Layers | 2 |
| | Discriminator Hidden Units | 500 |
| | Discriminator Non-linearity | Leaky ReLU |
| | Gradient Penalty ($\lambda$) | 10 |
| | Batch Size | 200 |
| | Early Stopping Patience | 5 |
| SurvivalGAN | Uncensoring Model | Survival Function Regression |
| | Time-to-event strategy | Survival Function |
| | Censoring Strategy | Random |
| | Dataloader Sampling Strategy | Imbalance Time Censoring |
| TVAE | No. Iterations | 1000 |
| | Batch Size | 200 |
| | Learning Rate | $1 \times 10^{-3}$ |
| | Weight Decay | $1 \times 10^{-5}$ |
| | Encoder No. Hidden Layers | 3 |
| | Encoder Hidden Units | 500 |
| | Encoder Non-linearity | Leaky ReLU |
| | Encoder Dropout Rate | 0.1 |
| | Decoder No. Hidden Layers | 3 |
| | Decoder Hidden Units | 500 |
| | Decoder Non-linearity | Leaky ReLU |
| | Decoder Dropout Rate | 0 |
| | Early Stopping Patience | 5 |
| | Data Encoder Max Clusters | 10 |
| | Embedding Width | 500 |
| TabDDPM | No. Iterations | 1000 |
| | Batch Size | 1024 |
| | Learning Rate | $2 \times 10^{-3}$ |
| | Weight Decay | $1 \times 10^{-4}$ |
| | No. of Time-Steps | 1000 |
| | Scheduler | Cosine |
| | Gaussian Loss Type | MSE |

# C    ADDITIONAL PERFORMANCE METRICS

Below we provide the comprehensive scores of all models evaluated in the paper. Table 12 presents the *covariate quality* and *downstream performance* metrics for all models assessed in Section 4.1. In Table 13, we report the *event-time distribution quality* metrics, including optimism, short-sightedness, and KM Divergence, for both conditional and unconditional models. Table 14 summarizes the results for the LLM experiment and Figure 4 shows Q-Q plots for the same, as discussed in Section 4.2. Table 15 and  16 summarizes the full results for the privacy experiment discussed in Section  4.4.

Table 12: Quality (JS Distance, WS Distance, and PVP) and downstream (C-Index and Brier Score) metrics. Models conditioning on $t$ and $e$ are highlighted (†), and Original is for the survival model trained on the real (training) data. Error bars are standard deviations for 5 repetitions.

| Metric | Method | AIDS | METABRIC | SUPPORT | GBSG | FLCHAIN |
|---|---|---|---|---|---|---|
| JS distance (↓) | SurvivalGAN | 0.013±0.00 | 0.009±0.00 | 0.008±0.00 | 0.008±0.00 | 0.009±0.00 |
| | TVAE† | 0.007±0.00 | 0.008±0.00 | **0.004±0.00** | 0.005±0.00 | 0.002±0.00 |
| | TabDDPM† | 0.007±0.00 | **0.007±0.00** | 0.013±0.00 | 0.005±0.00 | **0.001±0.00** |
| | CTGAN† | 0.013±0.00 | 0.020±0.01 | 0.005±0.00 | **0.003±0.00** | 0.004±0.00 |
| | ADS-GAN† | **0.006±0.00** | 0.009±0.00 | 0.005±0.00 | 0.004±0.00 | 0.010±0.01 |
| | TVAE | 0.011±0.00 | 0.009±0.00 | 0.007±0.00 | 0.007±0.00 | 0.003±0.00 |
| | DDPM | 0.006±0.00 | 0.007±0.00 | 0.006±0.00 | 0.005±0.00 | 0.002±0.00 |
| | CTGAN | 0.007±0.00 | 0.012±0.00 | 0.005±0.00 | 0.008±0.00 | 0.005±0.00 |
| | ADS-GAN | 0.006±0.00 | 0.007±0.00 | 0.007±0.00 | 0.005±0.00 | 0.005±0.00 |
| WS distance (↓) | SurvivalGAN | 0.112±0.01 | 0.039±0.00 | 0.043±0.00 | 0.019±0.00 | 0.052±0.00 |
| | TVAE† | **0.061±0.00** | **0.028±0.00** | **0.032±0.00** | 0.013±0.00 | **0.016±0.00** |
| | TabDDPM† | 0.159±0.02 | 0.089±0.00 | 0.308±0.02 | 0.056±0.00 | 0.028±0.00 |
| | CTGAN† | 0.095±0.00 | 0.133±0.01 | 0.034±0.00 | 0.013±0.00 | 0.019±0.00 |
| | ADS-GAN† | 0.082±0.00 | 0.037±0.00 | 0.036±0.00 | **0.011±0.00** | 0.018±0.00 |
| | TVAE | 0.075±0.00 | 0.031±0.00 | 0.037±0.00 | 0.013±0.00 | 0.017±0.00 |
| | DDPM | 0.079±0.00 | 0.031±0.00 | 0.049±0.00 | 0.015±0.00 | 0.016±0.00 |
| | CTGAN | 0.069±0.00 | 0.041±0.00 | 0.036±0.00 | 0.017±0.00 | 0.021±0.00 |
| | ADS-GAN | 0.065±0.00 | 0.035±0.00 | 0.038±0.00 | 0.013±0.00 | 0.017±0.00 |
| PVP (↓) | SurvivalGAN | 0.181±0.00 | 0.555±0.00 | 0.571±0.00 | 0.485±0.00 | 0.555±0.00 |
| | TVAE† | **0.090±0.00** | 0.444±0.00 | 0.457±0.06 | **0.142±0.00** | **0.222±0.04** |
| | TabDDPM† | 0.181±0.06 | 0.222±0.00 | 0.528±0.03 | 0.199±0.07 | **0.222±0.04** |
| | CTGAN† | 0.272±0.00 | 0.555±0.00 | 0.428±0.00 | 0.571±0.00 | 0.511±0.06 |
| | ADS-GAN† | 0.309±0.04 | 0.555±0.00 | 0.600±0.03 | 0.428±0.00 | 0.422±0.04 |
| | TVAE | 0.127±0.04 | 0.333±0.00 | 0.400±0.03 | 0.200±0.06 | 0.377±0.06 |
| | DDPM | 0.096±0.04 | **0.000±0.00** | 0.171±0.08 | 0.285±0.00 | 0.244±0.07 |
| | CTGAN | 0.181±0.00 | 0.555±0.00 | 0.428±0.03 | 0.285±0.00 | 0.444±0.00 |
| | ADS-GAN | 0.272±0.00 | 0.422±0.04 | 0.571±0.00 | 0.571±0.00 | 0.444±0.00 |
| C-Index (↑) | SurvivalGAN | 0.735±0.00 | 0.625±0.00 | 0.602±0.00 | 0.668±0.00 | 0.870±0.00 |
| | TVAE† | 0.737±0.00 | 0.612±0.00 | 0.583±0.00 | 0.672±0.00 | 0.872±0.00 |
| | TabDDPM† | 0.660±0.07 | 0.589±0.01 | 0.536±0.00 | 0.663±0.00 | 0.876±0.00 |
| | CTGAN† | 0.746±0.00 | 0.628±0.01 | 0.577±0.00 | 0.665±0.01 | 0.874±0.00 |
| | ADS-GAN† | **0.797±0.01** | **0.655±0.00** | 0.623±0.00 | **0.684±0.00** | **0.880±0.00** |
| | Original | 0.760±0.00 | 0.636±0.00 | 0.616±0.00 | 0.695±0.00 | 0.870±0.00 |
| | TVAE | 0.735±0.00 | 0.646±0.00 | 0.604±0.00 | 0.671±0.00 | 0.878±0.00 |
| | TabDDPM | 0.759±0.00 | 0.649±0.00 | **0.625±0.00** | 0.679±0.00 | 0.879±0.00 |
| | CTGAN | 0.779±0.00 | 0.647±0.00 | 0.606±0.00 | 0.679±0.00 | 0.878±0.00 |
| | ADS-GAN | 0.776±0.00 | 0.636±0.00 | 0.601±0.00 | 0.663±0.00 | 0.878±0.00 |
| Brier Score (↓) | SurvivalGAN | 0.068±0.00 | 0.205±0.00 | 0.202±0.00 | 0.212±0.00 | 0.096±0.00 |
| | TVAE† | **0.059±0.00** | 0.199±0.00 | 0.207±0.00 | 0.214±0.00 | 0.095±0.00 |
| | TabDDPM† | 0.063±0.00 | 0.212±0.00 | 0.217±0.00 | 0.215±0.00 | 0.096±0.00 |
| | CTGAN† | 0.061±0.00 | 0.199±0.00 | 0.205±0.00 | 0.215±0.01 | 0.089±0.00 |
| | ADS-GAN† | **0.059±0.00** | **0.197±0.00** | **0.198±0.00** | 0.213±0.00 | **0.084±0.00** |
| | Original | 0.062±0.00 | 0.200±0.00 | 0.195±0.00 | 0.205±0.00 | 0.095±0.00 |
| | TVAE | 0.061±0.00 | 0.204±0.00 | 0.206±0.00 | 0.210±0.00 | 0.093±0.00 |
| | DDPM | 0.060±0.00 | 0.200±0.00 | 0.199±0.00 | **0.207±0.00** | 0.087±0.00 |
| | CTGAN | 0.064±0.00 | 0.202±0.00 | 0.203±0.00 | 0.210±0.00 | 0.086±0.00 |
| | ADSGAN | 0.061±0.00 | 0.207±0.00 | 0.201±0.00 | 0.208±0.00 | 0.088±0.00 |

Table 13: Event-time distribution quality metrics. Models conditioning on $t$ and $e$ are highlighted (†). Error bars are standard deviations for 5 repetitions.

| Metric | Method | AIDS | METABRIC | SUPPORT | GBSG | FLCHAIN |
|---|---|---|---|---|---|---|
| Optimism ($\to 0$) | SurvivalGAN | 0.021±0.00 | 0.011±0.00 | 0.016±0.00 | 0.006±0.00 | 0.134±0.00 |
| | TVAE† | **0.000±0.00** | **0.000±0.00** | **0.000±0.00** | **0.003±0.00** | **0.001±0.00** |
| | DDPM† | **0.000±0.00** | **0.000±0.00** | **0.000±0.00** | **0.003±0.00** | **0.001±0.00** |
| | CTGAN† | **0.000±0.00** | **0.000±0.00** | **0.000±0.00** | **0.003±0.00** | **0.001±0.00** |
| | ADSGAN† | **0.000±0.00** | **0.000±0.00** | **0.000±0.00** | **0.003±0.00** | **0.001±0.00** |
| | TVAE | 0.023±0.00 | -0.003±0.00 | -0.014±0.00 | 0.004±0.00 | 0.022±0.00 |
| | DDPM | 0.021±0.00 | 0.001±0.00 | 0.001±0.00 | 0.026±0.00 | 0.005±0.00 |
| | CTGAN | -0.005±0.00 | 0.017±0.00 | -0.038±0.00 | 0.060±0.00 | -0.037±0.00 |
| | ADSGAN | 0.001±0.00 | -0.033±0.00 | -0.007±0.00 | 0.010±0.00 | 0.005±0.00 |
| Short Sightedness ($\to 0$) | SurvivalGAN | 0.007±0.00 | 0.124±0.00 | 0.020±0.00 | 0.019±0.00 | 0.005±0.00 |
| | TVAE† | **0.001±0.00** | **0.000±0.00** | **0.000±0.00** | **0.010±0.01** | **0.002±0.00** |
| | DDPM† | **0.001±0.00** | **0.000±0.00** | **0.000±0.00** | **0.010±0.01** | **0.002±0.00** |
| | CTGAN† | **0.001±0.00** | **0.000±0.00** | **0.000±0.00** | **0.010±0.01** | **0.002±0.00** |
| | ADS-GAN† | **0.001±0.00** | **0.000±0.00** | **0.000±0.00** | **0.010±0.01** | **0.002±0.00** |
| | TVAE | 0.058±0.00 | 0.148±0.00 | 0.002±0.00 | 0.017±0.00 | 0.018±0.00 |
| | DDPM | 0.002±0.00 | 0.000±0.00 | 0.002±0.00 | 0.015±0.00 | 0.003±0.00 |
| | CTGAN | 0.071±0.00 | 0.056±0.00 | 0.010±0.00 | 0.019±0.00 | 0.017±0.00 |
| | ADSGAN | 0.040±0.00 | 0.188±0.00 | 0.000±0.00 | 0.014±0.00 | 0.006±0.00 |
| KM Divergence ($\downarrow$) | SurvivalGAN | 0.021±0.00 | 0.082±0.00 | 0.064±0.00 | 0.049±0.00 | 0.134±0.00 |
| | TVAE† | **0.002±0.00** | **0.008±0.00** | **0.002±0.00** | **0.005±0.00** | **0.002±0.00** |
| | DDPM† | **0.002±0.00** | **0.008±0.00** | **0.002±0.00** | **0.005±0.00** | **0.002±0.00** |
| | CTGAN† | **0.002±0.00** | **0.008±0.00** | **0.002±0.00** | **0.005±0.00** | **0.002±0.00** |
| | ADS-GAN† | **0.002±0.00** | **0.008±0.00** | **0.002±0.00** | **0.005±0.00** | **0.002±0.00** |
| | TVAE | 0.031±0.00 | 0.042±0.00 | 0.025±0.00 | 0.027±0.00 | 0.031±0.00 |
| | DDPM | 0.021±0.00 | 0.019±0.00 | 0.011±0.00 | 0.026±0.00 | 0.007±0.00 |
| | CTGAN | 0.015±0.00 | 0.028±0.00 | 0.038±0.00 | 0.061±0.00 | 0.037±0.00 |
| | ADSGAN | 0.016±0.00 | 0.039±0.00 | 0.020±0.00 | 0.030±0.00 | 0.012±0.00 |

Table 14: Quality (JS distance, WS distance and PVP) and downstream (C-Index and Brier Score) metrics. Models conditioning on $t$ and $e$ are highlighted (†). BM refers to the best-performing model from Table 12.

| Dataset | Method | C-Index | Brier Score | JS distance | WS Distance | PVP |
|---|---|---|---|---|---|---|
| AIDS | SurvivalGAN | 0.735±0.00 | 0.068±0.01 | 0.013±0.00 | 0.12±0.00 | 0.181±0.00 |
| | GReaT† | 0.790±0.00 | 0.063±0.00 | **0.003±0.00** | **0.036±0.00** | **0.000±0.00** |
| | GReaT | 0.725±0.01 | 0.063±0.00 | 0.004±0.00 | 0.046±0.00 | 0.090±0.00 |
| | BM | **0.797±0.01** | **0.059±0.00** | 0.006±0.00 | 0.061±0.00 | 0.090±0.00 |
| METABRIC | SurvivalGAN | 0.625±0.00 | 0.205±0.00 | 0.009±0.00 | 0.039±0.00 | 0.555±0.00 |
| | GReaT† | 0.640±0.00 | **0.195±0.00** | **0.005±0.00** | **0.000±0.00** | **0.000±0.00** |
| | GReaT | 0.623±0.00 | 0.201±0.00 | 0.006±0.00 | **0.000±0.00** | 0.111±0.00 |
| | BM | **0.655±0.00** | 0.197±0.00 | 0.007±0.00 | 0.028±0.00 | **0.000±0.00** |
| SUPPORT | SurvivalGAN | 0.602±0.00 | 0.202±0.00 | 0.008±0.00 | 0.043±0.00 | 0.571±0.00 |
| | GReaT† | **0.630±0.00** | **0.198 ±0.00** | **0.002±0.00** | **0.000±0.00** | **0.071±0.00** |
| | GReaT | 0.627±0.00 | 0.200±0.00 | 0.003±0.00 | 0.020±0.00 | **0.071±0.00** |
| | BM | 0.625±0.00 | **0.198 ±0.00** | 0.004±0.00 | 0.032±0.00 | 0.171±0.08 |
| GBSG | SurvivalGAN | 0.668±0.00 | 0.212±0.00 | 0.008±0.00 | 0.019±0.00 | 0.485±0.00 |
| | GReaT† | **0.686±0.00** | **0.207±0.00** | 0.006±0.00 | 0.012±0.00 | **0.142±0.00** |
| | GReaT | 0.672±0.00 | **0.207±0.00** | 0.007±0.00 | **0.011±0.00** | **0.142±0.00** |
| | BM | 0.684±0.00 | **0.207±0.00** | **0.003±0.00** | **0.011±0.00** | **0.142±0.00** |
| FLCHAIN | SurvivalGAN | 0.870±0.00 | 0.096±0.00 | 0.009±0.00 | 0.052±0.00 | 0.555±0.00 |
| | GReaT† | **0.880±0.00** | **0.082±0.00** | **0.001±0.00** | **0.015±0.00** | **0.111±0.00** |
| | GReaT | 0.878±0.00 | 0.090±0.00 | **0.001±0.00** | 0.020±0.00 | 0.222±0.00 |
| | BM | **0.880±0.00** | 0.084±0.00 | **0.001±0.00** | 0.016±0.00 | 0.222±0.04 |

Table 15: Median value of Distance of closest record from the original. Models conditioning on $t$ and $e$ are highlighted † (our method). Error bars are standard deviations for 5 repetitions. The highest (best) values are in bold and the least (worst) values are underlined.

| Metric | Method | AIDS | METABRIC | SUPPORT | GBSG | FLCHAIN |
|--------|--------|------|----------|---------|------|---------|
| Median DCR | SurvivalGAN | 1.035±0.00 | 0.969±0.00 | 1.589±0.00 | 0.500±0.00 | **0.796±0.00** |
| | TVAE† | 0.883±0.00 | 0.877±0.00 | 1.511±0.00 | 0.476±0.00 | 0.642±0.00 |
| | TabDDPM† | **1.172±0.03** | 0.908±0.01 | 1.612±0.03 | 0.519±0.00 | 0.572±0.01 |
| | CTGAN† | 0.918±0.01 | 1.043±0.00 | 1.594±0.00 | **0.524±0.00** | 0.695±0.00 |
| | ADS-GAN† | 1.133±0.17 | 0.992±0.00 | **1.691±0.00** | 0.519±0.00 | 0.667±0.01 |
| | SMOTE | 0.388±0.00 | 0.698±0.00 | 0.958±0.00 | 0.290±0.00 | 0.381±0.00 |
| | TVAE | 1.044±0.02 | 0.813±0.00 | 1.405±0.00 | 0.432±0.00 | 0.553±0.00 |
| | TabDDPM | 1.020±0.01 | **1.087±0.00** | 1.611±0.00 | 0.477±0.00 | 0.567±0.00 |
| | CTGAN | 1.112±0.01 | 1.001±0.00 | 1.586±0.00 | 0.515±0.00 | 0.641±0.00 |
| | ADS-GAN | 1.158±0.01 | 0.945±0.00 | 1.666±0.00 | 0.475±0.00 | 0.533±0.00 |

Table 16: Minimum value of Distance of closest record from the original. Models conditioning on $t$ and $e$ are highlighted † (our method). Error bars are standard deviations for 5 repetitions. The highest (best) values are in bold and the least (worst) values are underlined.

| Metric | Method | AIDS | METABRIC | SUPPORT | GBSG | FLCHAIN |
|--------|--------|------|----------|---------|------|---------|
| Minimum DCR | SurvivalGAN | 0.048±0.00 | 0.172±0.00 | 0.326±0.00 | 0.062±0.00 | 0.057±0.00 |
| | TVAE† | 0.077±0.03 | 0.202±0.02 | 0.370±0.02 | 0.033±0.00 | 0.026±0.00 |
| | TabDDPM† | 0.095±0.00 | 0.193±0.05 | 0.403±0.01 | **0.065±0.00** | 0.037±0.00 |
| | CTGAN† | **0.139±0.01** | **0.215±0.01** | 0.321±0.01 | 0.045±0.01 | 0.054±0.01 |
| | ADS-GAN† | 0.102±0.01 | 0.185±0.04 | 0.391±0.01 | 0.053±0.01 | **0.066±0.02** |
| | SMOTE | 0.000±0.00 | 0.000±0.00 | 0.000±0.00 | 0.000±0.00 | 0.000±0.00 |
| | TVAE | 0.083±0.01 | 0.154±0.03 | 0.171±0.01 | 0.031±0.00 | 0.028±0.00 |
| | TabDDPM | 0.090±0.03 | 0.213±0.00 | 0.337±0.01 | 0.054±0.00 | 0.033±0.00 |
| | CTGAN | 0.109±0.02 | 0.194±0.02 | 0.316±0.02 | 0.046±0.01 | 0.024±0.00 |
| | ADS-GAN | 0.062±0.03 | 0.205±0.03 | **0.429±0.00** | 0.050±0.00 | 0.055±0.00 |

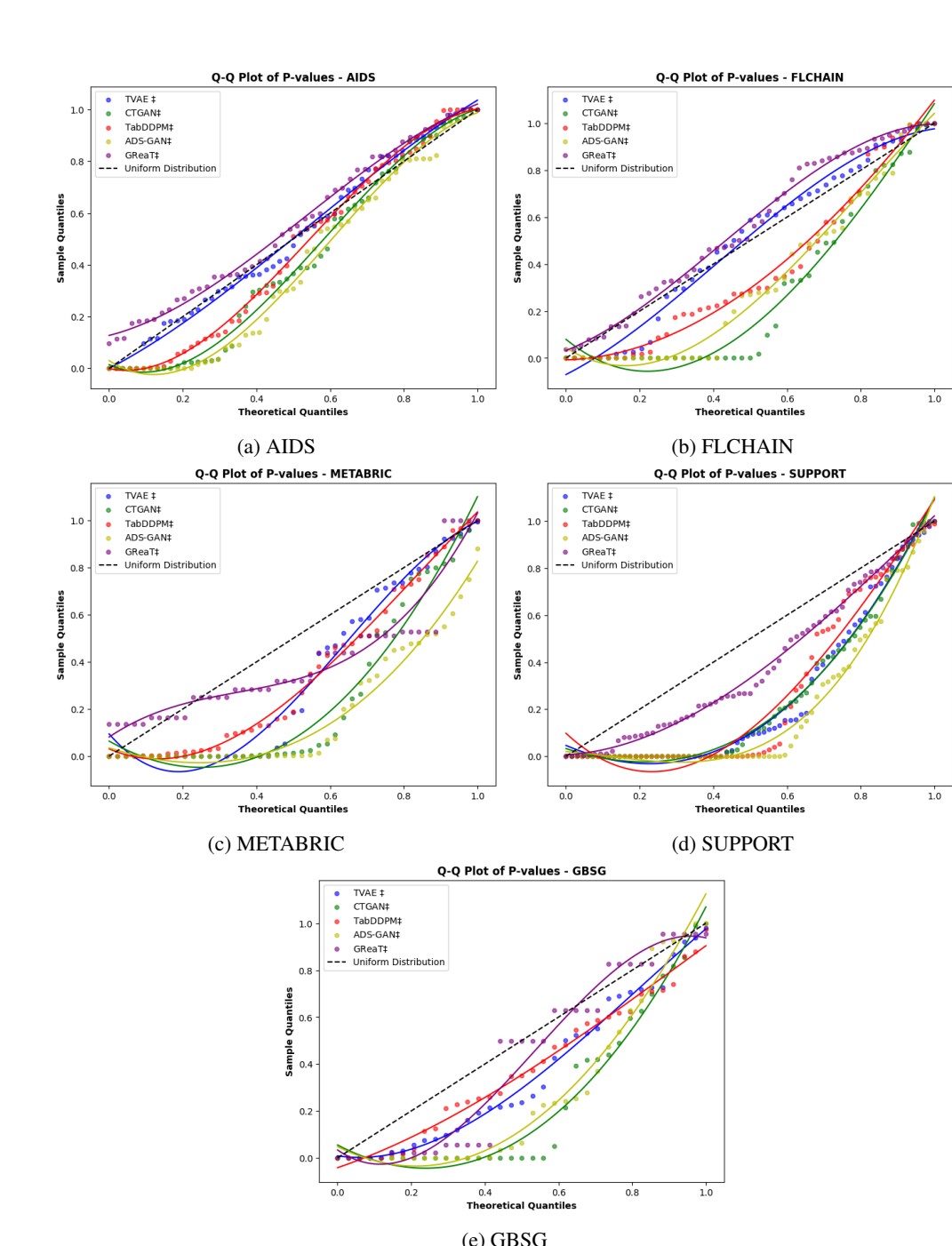

Figure 4: Q-Q plots comparing the $p$-value distributions of all conditional models (†) from Section 4.1 and 4.2. The dashed line represents the expected (uniform) distribution.

