# OpenReview forum: "Conditioning on Time is All You Need for Synthetic Survival Data Generation"
_ICLR.cc/2025/Conference — Submitted to ICLR 2025_

### Official Review · Reviewer_8iwG · 2024-10-19

**Soundness:** 1
**Presentation:** 2
**Contribution:** 1
**Rating:** 3
**Confidence:** 3

**Summary:**

Under the assumption of censoring at random in survival analysis, the authors propose to generate synthetic survival data by generating covariates conditioned on event time and type when using various existing conditional generative models (CTGAN, TAVE, ADS-GAN, TabDDPM) for tabular data.

**Strengths:**

1.	Any existing conditional generative models can be used based on the proposed idea.
2.	The authors present many experimental results.

**Weaknesses:**

1.	The setting in the paper is not clearly defined and remains vague. For example, the authors consider the right-censoring setting. However, it is necessary to specify which type of right-censoring is being considered. The authors mention that their method can be readily extended to left or interval censoring settings, but they do not explain how this extension would work.
2.	Typically, survival data is defined using survival time $t$, censoring time $C$, censoring indicator or type $e$, and covariates $X$. The authors do not include the censoring time $C$ in their setting. It is unclear whether their method generates the censoring time, and this should be clarified.
3.	The authors emphasize the simplicity of their method, yet they provide little explanation as to why this simple method would work intuitively. More interpretation is needed to justify this approach.
4.	The authors criticize the approximation error in covariates $x$ for compounding with the approximation error in $t$ for the conditional method in Norcliffe et al. (2023). However, the explanation is confusing as $\tilde{t}\sim p(t|x,e)$ is not conditional on $\tilde{x}$, so it is unclear how the approximation errors compound?
5.	Following weakness 4, it is unclear how the authors' method addresses the compounded approximation error issue. Based on (2), $\tilde{x}$ is drawn from a distribution conditional on $\tilde{t}$ and $\tilde{e}$. Approximation errors in $t$ and $e$ could still compounds with those for $x$.
6.	Using empirical distributions $p(t|e)$ and $p(e)$ could be very risky, since they may not be representative of the population (in many cases). How do the authors control the approximation error for small or unbalanced datasets? This is a challenge they emphasize in the introduction.
7.	The authors emphasis their method can accurately reproduce the real distribution of event times for both observed (uncensored) events and censored events. However, I do not see any theoretical support for this claim. Please show the “accurate reproduction” property. Please also clarify whether the “real distribution” refers to the marginal or joint distribution.
8.	The main idea is very close to that of Norcliffe et al. (2023), yet the authors do not clearly state the advantages of their approach. In survival analysis, it is crucial to generate informative $t$, but generating $t$ from $p(t|e)$ (the empirical version) is unlikely to be informative.
9.	Many statements in this paper are vague and unclear. For example, the description of using kernel density estimation in line 214-215 is vague. There are doubts of these statements. These types of statements should be avoided to ensure the accuracy of the work.
10.	Some notations can be confused. For example, in lines 153-154, the authors use $t_i$, $e_i$ without propor explanations. In line 070, the notation become $()_{n=1}^N$, which is inconsistent.
11.	The presentation of this paper needs improvement. For instance, the background paragraph introduces survival function, expected lifetime, etc.. But it is unclear how these are related to the authors' work or why they are included. Many sentences are long and awkward, such as the one in lines 197-198: "...which..., which...".
12.	The experiment section is merely a collection of results without clear interpretation. Many of the metrics used are not adequately introduced, while they are important to evaluate the proposed method, such as optimism and short-sightedness.

**Questions:**

1.	Please review the weaknesses.
2.	The Wasserstein distance (WS distance) for METABRIC in Norcliffe et al. (2023) is very different (much larger) than in your experiment. Could you explain the reasoning behind this difference?
3.	Some metrics in your evaluation are approximations, please clarify these approximations.

---

### Official Review · Reviewer_sCEG · 2024-10-21

**Soundness:** 2
**Presentation:** 3
**Contribution:** 2
**Rating:** 3
**Confidence:** 4

**Summary:**

This paper constructs a general method for generating synthetic survival data. Which consists of a triplet $(x, t, e)$, where $x$ are covariates, $e$ is an event and $t$ is the time the event happens. $e$ can represent censoring where we know the event of interest has not happened before $t$.

The main idea of the work is to factorize the joint distribution $p(x, t, e)$ as $p(x|t, e)p(t| e)p(e)$ where $p(t|e)p(e)$ is given by the empirical training data distribution and $p(x | t, e)$ is a generative model. The main claim of the paper is that this factorization is beneficial to generating survival data, and that this novelty is agnostic to what generative model $p(x |t, e)$ is used. This is in contrast to the main related work SurvivalGAN which factorizes the joint distribution as $p(t | x, e)p(x)p(e)$.

Extensive experiments are carried out on multiple real datasets and baselines, the results are mixed.

**Strengths:**

- The paper is nicely written, it is easy to follow.
- The main idea of conditioning on time is straightforward in a good way, it is also quite general by being agnostic to the generative model.
- The experiments are detailed and extensive. Many models and real datasets are tested. Both positive and negative results are given which is appreciated.

**Weaknesses:**

The main weakness is the evaluation and experiments run. I appreciate a lot of work has gone into generating the existing results, but I do not think they match the claims of the paper, and that there are better comparisons to be made. The main claim is that conditioning on time will lead to performance increase and is agnostic to the generative model. In this case the more convincing result will be to test many generative models, each with different factorizations: the proposed one $p(x | t, e)p(t| e)p(e)$, the SurvivalGAN one $p(t |x , e)p(x)p(e)$ and the unconditional one $p(x, t, e)$. Then the most convincing result will be that if for each model the best performance is seen with the proposed factorization. Tables like this are given in the Appendix, tables 12, 13, 14, 15, 16. In these tables the proposed and unconditional distributions are tested, and the results are quite mixed and not convincing that this factorization really improves the generated synthetic data.

Currently the main results presented in tables 1, 2, 3, 4, 5 compare SurvivalGAN and unconditional models to multiple baselines with the proposed factorization with four separate generative models. Effectively this gives the proposed method 4 chances to achieve the best result. The way to make this more rigorous is to test SurvivalGAN with multiple different generative models, to show that in each case the proposed factorization is better. Which is equivalent to the above.

Ultimately the evaluation currently is the weak point of the paper. I like the idea and its simplicity, however, since (as far as I can tell) there is very little theory to support this factorization over others, then the empirical evaluation must be outstanding and currently it is not convincing enough. The paper would be most improved with this type of evaluation, testing multiple generative models (as has been done), but comparing across the different factorizations for each one: 1) the proposed factorization $p(x | t, e)p(t| e)p(e)$, 2) the SurvivalGAN style factorization $p(t |x , e)p(x)p(e)$ and 3) an unconditional one $p(x, t, e)$. If in this experiment the proposed factorization is consistently best regardless of dataset or generative model then this is a far more convincing result. Currently Table 13 demonstrates this a lot more than the others, showing that this factorization can work well. The reason is that Table 13 is evaluating the $p(t, e)$ distribution and the proposed method uses the empirical train set for this, so is going to be best, but if downstream performance supports this then that shows that this distribution is key to generate good survival data, and then suports the method more strongly.

**Questions:**

See weaknesses.

---

### Official Review · Reviewer_pobY · 2024-11-02

**Soundness:** 1
**Presentation:** 2
**Contribution:** 1
**Rating:** 3
**Confidence:** 5

**Summary:**

This paper presents a synthetic data generation method that aims to provide a simpler alternative to established methods like SurvivalGAN, but it suffers from significant theoretical and practical flaws that undermine its contribution. The method lacks a clear causal motivation, relies on questionable assumptions, and introduces privacy risks. Its empirical results are unimpressive, and the supposed benefits over existing methods are either marginal or unsubstantiated. Overall, while the paper proposes a straightforward approach, the methodology lacks the rigor and depth needed to be a meaningful advancement in synthetic data generation for survival analysis.

In summary, while the paper presents a straightforward approach to synthetic data generation, it is marred by theoretical inconsistencies, privacy concerns, and underwhelming empirical results. The lack of alignment between assumptions and methodology, combined with the inadequate validation and privacy assessment, suggests that the approach is neither a substantial nor a practical improvement over existing methods.

**Strengths:**

The paper introduces a synthetic data generation approach that prioritizes simplicity and ease of implementation, offering a straightforward alternative to more complex methods like SurvivalGAN.

**Weaknesses:**

Below is a list of weaknesses.
1. The sequence of modeling in this paper—generating from p(e), p(t∣e), and p(x∣t,e)—is poorly motivated and lacks a sound theoretical basis. Conditioning covariates on both time and event does not align with standard causal structures, where covariates would typically precede and influence both event and time. This choice suggests a misunderstanding of causal dependencies, undermining the model's credibility. Without a more thoughtful theoretical foundation, the proposed structure appears arbitrary and counterintuitive.

2. Additionally, the generation method is overly simplistic, involving direct sampling from the empirical distribution for p(t) and p(e) and applying any conditional generative model for p(x∣t,e). While this approach may seem appealing due to its simplicity, it fails to capitalize on more sophisticated methods that could enhance performance or generalizability. Direct sampling from the empirical distribution introduces serious privacy concerns. Each synthetic sample directly mirrors the time and event distributions of real data, making this approach antithetical to the purpose of synthetic data, which is to offer privacy-preserving alternatives to real data. Although the authors employ the DCR metric to assess privacy, this measure lacks theoretical guarantees and is incompatible with robust privacy frameworks like differential privacy. Furthermore, while the authors acknowledge that kernel density estimation could reduce privacy risks, they do not explore this option experimentally, leaving questions about the practical utility and safety of their approach.

3. The authors’ assumption that "observed and censoring times are conditionally independent given the covariates" is also problematic. Their generation process contradicts this assumption. If t⊥⊥e∣x, then they should be able to generate from both marginals independently. For instance, if the conditional generator of x given t and e is constant, then t and e will not be conditionally independent, thus breaking a fundamental assumption in survival analysis. This inconsistency between theoretical assumptions and practical implementation significantly weakens the methodological foundation of the paper.

4. Building on the above, this assumption that observed and censoring times are conditionally independent given the covariates, raises significant concerns in real-world settings, where this assumption is often unrealistic, as censoring times can be linked to factors that influence survival times even when conditioned on covariates. For instance, in medical studies, a patient’s decision to exit a study (censoring) might be directly related to their declining health—a factor that also impacts survival—thus violating the assumption of conditional independence. Additionally, this assumption does not account for informative censoring, where censoring itself is indicative of an imminent event, such as a severe health deterioration. Ignoring this dependency can lead to substantial bias, as informative censoring demands adjustments beyond the simple conditional independence framework. Furthermore, the assumption introduces potential biases in models that involve complex covariate interactions, as it overlooks latent factors affecting both censoring and survival. This is especially problematic in high-dimensional settings, where covariate interactions are often intricate and less observable. By assuming independence, the model also loses flexibility in capturing dependencies within the joint distribution of censoring and survival times, leading to suboptimal performance when censoring mechanisms interact with the event in ways that covariates alone cannot account for. Finally, the assumption is difficult to validate, as it is practically impossible to confirm whether censored individuals would have different survival times. This lack of verifiability undermines the robustness of the analysis and may lead to overly optimistic model performance if the assumption does not hold in the underlying data. Together, these issues suggest that this assumption, while convenient, significantly limits the reliability and practical applicability of the method presented in the paper.

5. The empirical results presented in this paper do little to substantiate the method’s efficacy. The improvements obtained through conditional sampling are marginal, with slim gains across covariate quality metrics and Brier scores, which calls into question the practical impact of this approach. For a method that claims to advance synthetic data generation, these results are underwhelming, as they demonstrate limited progress over simpler baselines. Additionally, the performance of the large language model (LLM) used for generation does not show a substantial advantage over non-LLM methods, and its generation time is many orders of magnitude slower. The time cost far outweighs any minor gains in performance, making this approach inefficient and unsuitable for practical use.

6. Experiment 4.3, which highlights the supposed benefits of the proposed method for subgroup analysis, is also unclear. The authors do not adequately define what constitutes good performance for subgroup analysis, leaving the results ambiguous. Interestingly, the findings suggest that SurvivalGAN achieves more balanced downstream predictions across different racial groups, which could be seen as a point in favor of SurvivalGAN rather than the proposed approach.

7. Importantly and totally ignored by this paper, SurvivalGAN offers a variety of performance metrics specifically tailored to synthetic data validation, whereas this paper relies on less appropriate metrics and a flawed validation mechanism. This lack of rigor in validation only further undermines the credibility of the proposed method.

**Questions:**

1. Could the authors clarify how their generation process aligns with the stated assumption of conditional independence between observed and censoring times given the covariates? Additionally, could they provide theoretical or empirical evidence demonstrating that their method preserves this property? This clarification may help address the apparent inconsistency between their assumptions and implementation, and potentially refine the methodological foundation of the paper.
2. Could the authors conduct additional experiments to better highlight the strengths of their method, highlighting all the various metrics, including those in SurvivalGAN? Additionally, could they provide a more in-depth analysis of the trade-offs between performance gains and computational costs, especially given that the large language model (LLM) used for generation shows only marginal improvements over non-LLM methods while incurring significantly higher generation times?
3. Could the authors provide a more detailed discussion of how their method compares to SurvivalGAN in terms of balancing predictions across different racial groups, particularly given that SurvivalGAN appears to achieve more consistent results across subgroups?

---

### Meta-Review · Area_Chair_vQTj · 2024-12-21

**Metareview:**

The paper proposes a synthetic data generation method for survival analysis by factorizing the joint distribution


Strengths:

+ Implementable approach that prioritizes simplicity


Weaknesses:

+ The proposed causal structure conflicts with standard assumptions in survival analysis where covariates typically precede and influence events/times

+ Empirical results shows only marginal improvements over baselines while introducing significant computational overhead with LLM-based generation

+ Direct sampling from empirical distributions makes synthetic data vulnerable to privacy breaches without proper safeguards

**Additional Comments On Reviewer Discussion:**

The reviewers are in agreement that the paper does not meet the threshold of acceptance.

---

### Decision · Program_Chairs · 2025-01-22

Reject